# Cement augmentation for trochanteric femur fractures: A meta-analysis of randomized clinical trials and observational studies

Ingmar F. Rompen[1][*], Matthias Knobe[1], Bjoern-Christian Link[1], Frank J. P. Beeres[1], Ralf Baumgaertner[1], Nadine Diwersi[1], Filippo Migliorini[2], Sven Nebelung[3], Reto Babst[4], Bryan J. M. van de Wall[1,4]

1 Department of Orthopaedic and Trauma Surgery, Lucerne Cantonal Hospital, Lucerne, Switzerland, 2 Department of Orthopaedics, RWTH Aachen University Hospital, Aachen, Germany, 3 Department of Radiology, RWTH Aachen University Hospital, Aachen, Germany, 4 Department of Health Science and Medicine, University of Lucerne, Lucerne, Switzerland

☯ These authors contributed equally to this work.
* rompen@me.com

**Data Availability Statement:** All relevant data are within the paper and its Supporting Information files.

## Abstract

### Introduction

To date, it is unclear what the clinical benefit of cement augmentation in fixation for trochanteric fractures is. The aim of this meta-analysis is to compare cement augmentation to no augmentation in fixation of trochanteric femur fractures in the elderly patients (>65 years) following low energy trauma.

### Methods

PubMed/Medline/Embase/CENTRAL/CINAHL were searched for both randomized clinical trials (RCT) and observational studies comparing both treatments. Effect estimates were pooled across studies using random effects models. Subgroup analysis was performed stratified by study design (RCTs and observational studies). The primary outcome is overall complication rate. Secondary outcomes include re-operation rate, mortality, operation duration, hospital stay, general quality of life, radiologic measures and functional hip scores.

### Results

A total of four RCT's (437 patients) and three observational studies (293 patients) were included. The effect estimates of RCTs were equal to those obtained from observational studies. Cement augmentation has a significantly lower overall complication rate (28.3% versus 47.2%) with an odds ratio (OR) of 0.3 (95%CI 0.1–0.7). The occurrence of device/fracture related complications was the largest contributing factor to this higher overall complication rate in the non-augmented group (19.9% versus 6.0%, OR 0.2, 95%CI 0.1–0.6). Cement augmentation also carries a lower risk for re-interventions (OR 0.2, 95%CI 0.1–0.7) and shortens the hospital stay with 2 days (95%CI -2.2 to -0.5 days). The mean operation time was 7 minutes longer in the augmented group (95%CI 1.3–12.9). Radiological scores

**Funding:** The authors received no specific funding for this work.

**Competing interests:** The authors have declared that no competing interests exist.

(lag screw/blade sliding mean difference -3.1mm, 95%CI -4.6 to -1.7, varus deviation mean difference -6.15˚, 95%CI; -7.4 to -4.9) and functional scores (standardized mean difference 0.31, 95%CI 0.0–0.6) were in favor of cement augmentation. Mortality was equal in both groups (OR 0.7, 95%CI 0.4–1.3) and cement related complications were rare.

## Conclusion

Cement augmentation in fixation of trochanteric femoral fractures leads to fewer complications, re-operations and shorter hospital stay at the expense of a slightly longer operation duration. Cementation related complications occur rarely and mortality is equal between treatment groups. Based on these results, cement augmentation should be considered for trochanteric fractures in elderly patients.

## Introduction

Trochanteric femur fractures are a major health problem in the elderly population. It is estimated that around 1.5 million people per year worldwide suffer from hip fractures with rising numbers due to aging of the population [1].

Treatment of choice in trochanteric femur fractures is osteosynthesis with intramedullary nailing devices such as Gamma3 nail, TFNA, PFNA or sliding/dynamic hip screw systems (SHS/DHS) [2, 3]. It is, however, still associated with a mechanical complication rate up to 20% despite modifications and improvements of osteosynthetic devices [4]. Failure is mainly caused by varus collaps and cut-out of the implant [5]. It is thought that rotational head moments combined with migration and femoral neck shortening precede these complications [6–9]. A solution to this problem, especially in osteoporotic bone, might be cement augmentation. Biomechanical studies have shown that it increases the resistance of the osteosynthesis device to the shear stress that comes about during the load, preserving the implant from the aforementioned complications, especially in cases of eccentric implant position or low bone density [10, 11].

To date there is no clear evidence suggesting a clinical benefit of cement augmentation. Individual studies have either failed to show a significant difference or found small differences [12–15]. A formal meta-analysis on this topic has not been previously published.

The aim of this meta-analysis is to compare cement augmentation with no augmentation in fixation of trochanteric femur fractures in elderly patients (>65 years) following a low energy trauma. The primary outcome is overall complication rate. Secondary outcomes include re-operation rate, mortality, operation duration, hospital stay, radiological measures and functional hip scores as well as general quality of life. To evaluate these factors, both randomized controlled trials and observational studies were included.

## Methods

This meta-analysis was performed according to the Preferred Reporting Items for Systematic Reviews and Meta-analysis (PRISMA) checklist and the Meta-Analysis of Observational Studies in Epidemiology (MOOSE) [16, 17]. We applied a standardized method employed in all meta-analysis of our studygroup [18–20]. Ethical approval was not required.

## Search strategy and selection criteria

We performed a comprehensive search of electronic databases (PubMed, Embase, CENTRAL and CINAHL) for studies on cement augmentation for trochanteric fractures. S1 Table in S1 File describes the full search synthax. The search was performed on July 4, 2020.

All randomized controlled trials and observational studies that compared cement augmentation with no augmentation in fixation for trochanteric femur fractures in elderly patients (>65 years) following a low energy trauma were included in this review. Devices used for fixation included TFNA (trochanteric fixation nail advanced, DePuy-Synthes®), PFNA (proximal femoral nail antirotation, DePuy-Synthes®), Gamma3 nail (Stryker®) and SHS/DHS (sliding/dynamic hip screw, Stratec®, DePuy-Synthes®). Other inclusion criteria included minimal follow-up duration of 6 months, reporting on the outcomes of interest and availability of full-text.

Exclusion criteria were cadaveric studies, studies on pathologic fractures, case reports, languages other than English, Dutch, French, German, Spanish or Italian. Studies using other devices than mentioned above were excluded due to the inability for cement augmentation or the infrequency of usage in modern treatment of trochanteric fractures [3].

Two reviewers assessed the search and the inclusion of studies independently (IFR, BJMvdW). Disagreement was solved by consensus with a third reviewer (FJPB).

## Data extraction

Study and patient characteristics were collected in a predefined data extraction sheet and included: first author, publication year, study period and country in which study was performed, design of the study, study population size, type of cement and type of implant used. Furthermore, we extracted the type of fracture (using the AO/OTA-classification), gender, reduction quality, lag screw/blade position, the patient's history of smoking or diabetes, and follow up duration [21].

## Quality assessment

The same two reviewers (IFR, BJMvdW) assessed the methodological quality of included studies independently using the MINORS-Criteria (Methodological Index for Non-Randomized Studies) [22]. Disagreement was resolved by consensus. Details are described in S2 Table in S1 File.

## Primary outcome

The primary outcome of interest was the overall complication rate in both groups. Additionally, complications were subdivided in fracture/implant related, systemic and thromboembolic complications.

Fracture/implant related complications included malunion, non-union, implant bending or breakage, superficial wound infections as well as deep wound infections, cement leakage, refracture of the operated hip, irritation of the iliotibial band due to lag screw/blade sliding, postoperative hematoma, and extrusion of the lag screw/blade. Cutting of the head-neck element included both cut-through (central perforation of the lag screw/blade into the hip joint, without any displacement of the head-neck fragment) and cut-out (deviation of the neck-shaft angle into varus leading to extrusion of the screw from the femoral head) [23, 24].

Systemic complications encompassed delirium, pneumonia, cerebral strokes, myocardial infarction, renal insufficiency and, major bleeding in other locations than the operation site and bone cement implantation syndrome (BCIS). BCIS is a rare adverse event during a

procedure using cement augmentation and is characterized by hypoxia, hypotension, and/or unexpected loss of consciousness [25].

Thromboembolic complications included all venous thrombembolisms in the follow-up period and are listed as a part of systemic complication as well as separately [12, 14].

### Secondary outcomes

Secondary outcomes included re-interventions, mortality, time-to-union, hospital stay, operation duration, radiological outcomes, functional hip scores, visual analogue scale (VAS) for pain, and general quality of life measured at 6 to 12 months after the operation.

Re-interventions included all re-operations performed on the affected bone/fracture site during follow-up.

Radiological outcomes included sliding of the screw/blade in millimeters (mm) in the anteroposterior (AP) X-ray and varus deviation in degrees also using the AP radiograph.

The results of functional hip scores and general quality of life scores were standardized and pooled. Scoring systems included the Harris Hip Score as well as the Parker mobility score for functional results and the Bartel-Index for general quality of life.

### Statistical analysis

Continuous variables were presented as means with standard deviation (SD) or range. If required information was converted to mean and SD using the methods described in the Cochrane Handbook for Systematic Reviews of Interventions. Dichotomous variables were presented as counts and percentages. Effects of treatment options on continuous outcomes were pooled using the (random effects) inverse variance weighting method. They were presented as mean difference (radiological scores) or standardized mean difference (functional hip and general quality of life scores) with corresponding 95% confidence interval (95%CI). Binary outcomes were analysed using the (random effects) Mantel-Haenszel method. They were presented as odds ratio (OR), risk difference (RD), mean difference (MD) and standardized mean difference (SMD) with a 95% confidence interval (95%CI). Hereafter the terms weighted OR, weighted RD, weighted MD and weighted SMD are used for brevity.

Heterogeneity between studies was quantified by the I2 statistic and assessed for all OR's by visual inspection of forest plots. All analyses were stratified according to study design (randomized clinical trials versus observational studies). Differences between the pooled estimates of both study designs were assessed using the $\chi$2-test. The threshold for significance was set at a p-value of 0.05. All funnel plots of each analyses can be found in the (S10–S17 Figs in S1 File). Review Manager (RevMan, version 5.4) was used for all statistical analysis.

### Sensitivity analysis

Sensitivity analyses were performed for the primary outcomes on high quality studies, type of cement used (PMMA versus calciumphosphate) and type of implant (sliding hip screws versus cephalomedullary nailing devices). High quality studies were defined as studies with a MINORS score of 19 or higher (range 0–24).

## Results

### Literature search

A total of 1818 references were evaluated. A detailed description of the search and screening is shown in Fig 1. Finally three observational studies [13, 14, 26] and four randomized controlled trials [12, 15, 27, 28] fulfilled the criteria.

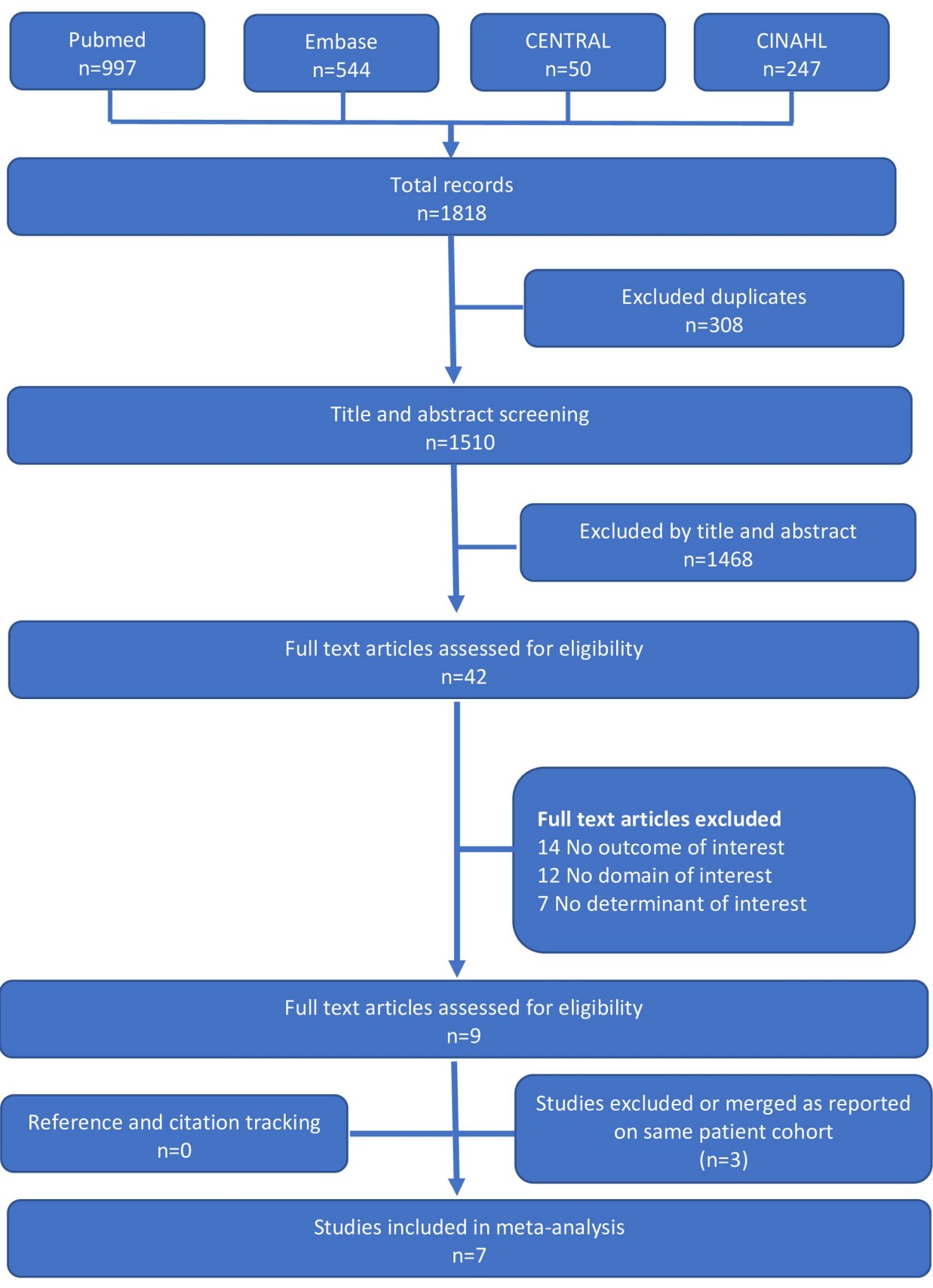

**Fig 1.**

## Baseline study characteristics

The seven studies included a total number of 730 patients of which 369 received cement augmentation following fixation and in 361 no augmentation was conducted (Table 1). All baseline characteristics including age, gender, ASA, diabetes, smoking history, AO classification, type of cement, blade position, reduction quality are described in Table 1 and S4 Table in S1 File. All characteristics were equally distributed among treatment groups.

## Quality assessment

The mean quality of all studies was 19 points (range 17–21) using the MINOR-Criteria [22]. For randomized controlled studies the mean was 19.5 (range 17–21) and for observational studies the mean was 18.3 (range 18–19). Details can be seen in S3 Table in S1 File.

## Primary outcomes—complications

Primary outcome defined as overall complications were reported in five studies; two randomized clinical trials and three observational studies [12–14, 26, 27]. The overall complication rate was significantly lower in the augmented group (28.3% versus 47.2%) with a weighted OR of 0.3 (95%CI 0.1–0.7, I2: 75%) (Fig 2). All complications per treatment group are listed in the S6 Table in S1 File.

The occurrence of device/fracture related complications was the largest contributing factor to the higher overall complication rate in the non-augmented group (19.9% versus 6.0%, OR 0.2, 95%CI 0.1–0.6, I2:53%) (S1 Fig in S1 File). Systemic complications occurred at an equal rate in both groups (OR 0.67 95%CI 0.3–1.6, I2:69%) (S2 Fig in S1 File).

No statistical significant difference was detected in the occurrence of thromboembolic events with events occurring in 3.9% in the augmented versus 0.4% in the non-augmented group (OR 6.0, 95%CI 1.0–35.6, I2:0%) (S3 Fig in S1 File).

**Table 1. Baseline characteristics.**

| Author | Year | Country | Study design | Study period | Device | Type of cement | Eligible Number of Patients | | Gender (female/male) | | Mean Age (SD) | | AO31-(A1/A2/A3) | | ASA (I/II/III/IV) | | Follow up |
|---|---|---|---|---|---|---|---|---|---|---|---|---|---|---|---|---|---|
| | | | | | | | augmented | control | augmented | control | augmented | control | Augmented | control | augmented | control | |
| *RCT* | | | | | | | | | | | | | | | | | |
| Kammerlander | 2018 | Germany | RCT | 2012–2015 | PFNA | PMMA | 105 | 118 | 87/18 | 99/19 | 86.1 (4.6) | 85.6 (4.9) | 0/96/9 | 0/96/22 | 10/31/59/4 | 13/44/55/5 | 12 months |
| Dall Oca | 2010 | Italy | RCT | 2006–2010 | Gamma3 nail | PMMA | 40 | 40 | 26/14 | 30/10 | 85.3 (2.3) | 82.3 (1.2) | 0/20/15 (n = 35) | 0/22/14 (n = 36) | nr | nr | 12 months |
| Lee | 2009 | Taiwan | RCT | 2005–2007 | DHS | PMMA | 55 | 53 | 30/25 | 29/24 | 82.6 (4.9) | 81.3 (5.8) | 0/46/9 | 0/45/8 | 6/26/23/0 | 9/23/21/0 | 14 months |
| Mattson | 2004 | Sweden | RCT | nr | DHS | calcium-phosphate | 14 | 12 | 12/2 | 10/2 | 83.7 (7.25) | 81.7 (7.25) | nr | nr | nr | nr | 6 months |
| *Observational studies* | | | | | | | | | | | | | | | | | |
| Yee | 2020 | China | OS | 2015–2019 | TFNA | PMMA | 47 | 29 | 39/8 | 23/6 | 85.1 (7.4) | 86.1 (7.7) | 7/28/12 | 3/14/12 | 0/15/32/0 | 0/9/20/0 | 12 months |
| Kulachote | 2019 | Thailand | OS | 2010–2017 | PFNA | PMMA | 68 | 67 | 55/13 | 44/23 | 85 (6) | 83 (6) | 19/43/6 | 11/54/2 | 0/0/28/40 | 0/0/40/27 | 12 months |
| Kim | 2018 | South Korea | OS | 2014–2017 | PFN | calcium-phosphate | 40 | 42 | 25/15 | 24/18 | 81.6 (16.3) | 82.3 (14.2) | 0/35/5 | 0/36/6 | nr | nr | 6 months |

PMMA: polymethylmethacrylate.

RCT: randomised clinical trial.

OS: Observational study.

nr: not reported.

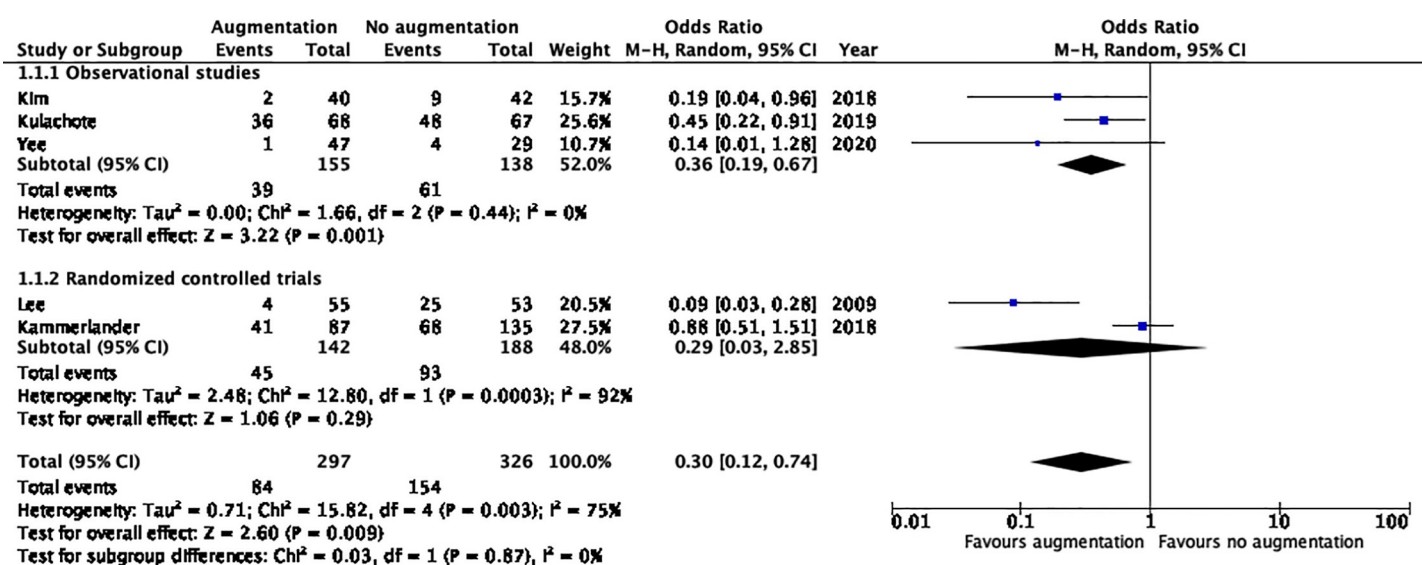

**Fig 2. Overall complications.**

There was no difference in pooled risk estimates between randomized clinical trials and observational studies.

## Secondary outcomes

### Re-intervention

Five studies reported on re-intervention [12, 13, 15, 26, 27]. Re-intervention was required less often in the augmented group (1.6% versus 7.4%, OR 0.2, 95%CI 0.1–0.7, I2:0%, Fig 3). All indications for re-intervention are listed separately in the S7 Table in S1 File.

There was no difference in pooled risk estimates between randomized clinical trials and observational studies.

### Mortality

Three studies reported a one year mortality [12, 14, 28]. In one paper, only three-month mortality was available [26]. For simplicity, these measures were pooled. Mortality in the

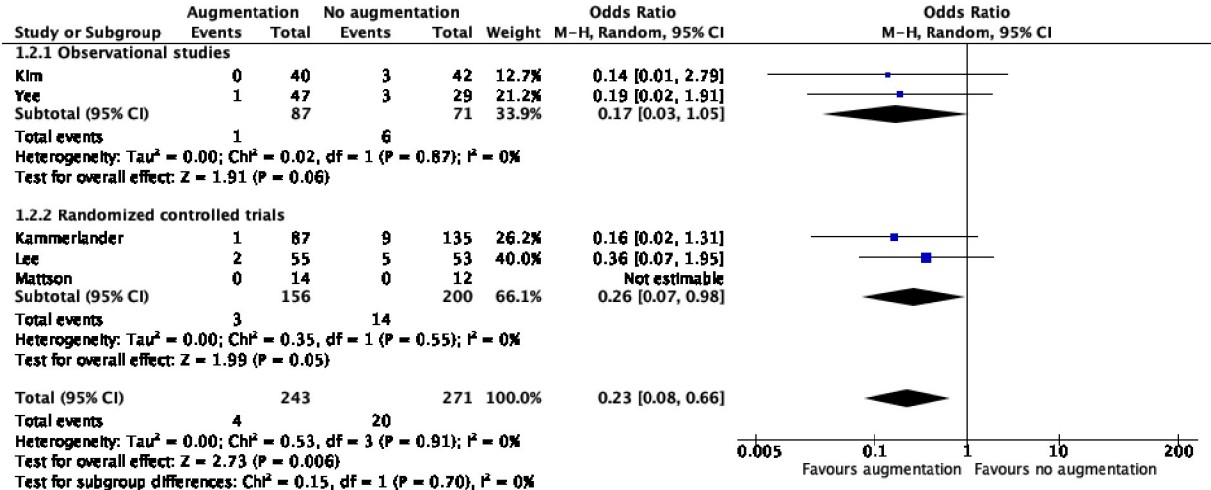

**Fig 3. Re-interventions.**

augmented group was 9.2%, versus 11.8% in the non-augmented group. There was no significant difference in postoperative mortality between both treatment groups (OR 0.7, 95%CI 0.4–1.3, I2:0%, S4 Fig in S1 File).

The pooled risk estimates of randomized clinical trials and observational studies were equal.

## Hospital stay

The duration of hospital stay was reported in two observational studies and two randomized controlled trials and was 1.9 days shorter in the augmented group (95%CI -2.2–0.5, I2:0%, S5 Fig in S1 File) [13, 15–17]. There was no difference between the pooled estimates of randomized clinical trials and observational studies.

## Operation duration

Four studies reported on operation duration–two randomized clinical trials and observational studies. The mean operation time was 7 minutes longer in the augmented group (95%CI 1.3–12.9, I2:95%, S6 Fig in S1 File) [13, 14, 27, 28].

## Time-to-union

Time to union was reported in two studies [14, 27]. However, only one study reported this measure for both treatment groups separately without any significant difference: 12.9 weeks (SD 3.1) for augmented versus 12.5 weeks (SD 1.6) for non-augmented devices) [14].

## Radiological outcomes

In five studies it was possible to calculate the amount of sliding of the lag screw/blade in AP view at 6–12 months follow-up [12–14, 27, 28]. There was significantly less sliding of the screw/blade in the augmented group (MD -3.1mm, 95%CI -4.6 to -1.7, P<0.0001, S7 Fig in S1 File).

Varus deviation in degrees was measured in an AP X-ray 6–12 months after surgery in three studies [13, 15, 27]. Significantly less varus deviation was observed in the augmented group (MD -6.1 degrees, 95%CI: -7.6 to -4.9, S8 Fig in S1 File).

## Functional hip scores

Four studies reported on functional hip scores measured 6–12 months after surgery: One observational study and three RCTs respectively [12, 13, 27, 28]. There was a significant difference in pooled postoperative scores (SMD -0.3, 95%CI -0.6–0.0, Fig 4) favoring augmentation. There was no difference between the pooled estimates of RCTs and observational studies.

In addition, Kulachote et al. reported return to pre-ambulatory setting. In the augmented group 48% of the patients returned to their level of pre-fracture mobility whereas only 29% of the non-augmented patients did (p = 0.43) [14].

## General quality of life

A general quality of Life scores was only described in one study using the Barthel-Index [12]. There was no significant difference in scores measured at 12 months follow-up.

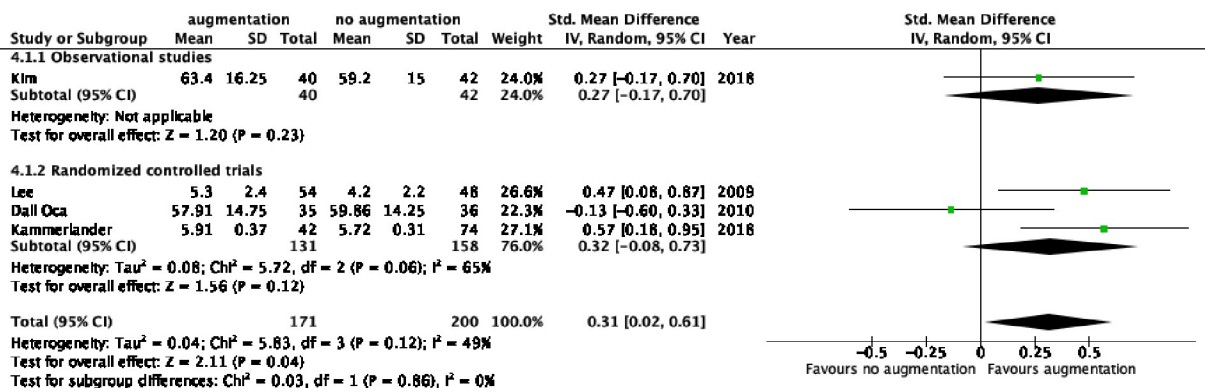

**Fig 4. Functional hip scores follow up.**

## Visual analogue scale for pain

Visual analogue scale (VAS) for pain was reported in two studies 6 and 12 months postoperatively [13, 27]. Significantly less pain was observed in the augmented group (MD -0.5pts. 95CI -0.8- -0.3, I2:0%) (S9 Fig in S1 File).

## Sensitivity analysis

Table 2 shows the results of the sensitivity analysis with regard to high quality studies, SHS versus cephalomedullary nailing devices and for type of cement used. No significant differences were found between the main analyses and these sensitivity analyses.

## Discussion

This meta-analysis of both randomized clinical trials as well as observational studies compared cement augmentation to no augmentation in fixation of trochanteric femur fractures in elderly patients. Cement augmentation leads to considerably fewer peri- and postoperative complications (28.3% versus 47.2%), implant related complications (6% versus 19.9%) and re-interventions (1.6% versus 7.4%).

Additionally, it demonstrated better radiological results (less sliding of the screw/blade and varus deviation), better functional results regarding mobility, less hip pain and a shorter hospital stay. This was however at the expense of a longer operation duration. There was no significant difference in mortality between the two treatment groups in the reported follow-up

**Table 2. Sensitivity analysis.**

| Type of studies | OR | 95%CI |
|---|---|---|
| **Overall** | 0.3 | 0.12–0.74 |
| **High quality** | 0.48 | 0.09–2.67 |
| **Lower quality** | 0.21 | 0.07–0.63 |
| **PMMA cement** | 0.32 | 0.11–0.89 |
| **Calciumphosphate cement** | 0.19 | 0.04–0.96 |
| **Cepholomedullary devices** | 0.47 | 0.23–0.97 |
| **SHS**/DHS | 0.09 | 0.03–0.28 |

PMMA = Polymethylmethacrylate.
SHS/DHS = Sliding- /Dynamic hip screw.

period. Randomized controlled trials and observational studies showed the same effect estimates in all analyses.

## Comparison with literature

Only one systematic review has been published on this topic in 2013 [29]. Comparable to our study, they found a lower incidence of overall complications and better radiological results in the cement augmentation group. The main difference with the present meta-analysis is that the previous mostly focused on SHS. We analyzed both SHS and cephalomedullary devices combined and performed sensitivity analysis to investigate whether there was a difference in results between both implants. Furthermore, we had four additional studies at our disposal increasing sample size and power of the present meta-analysis [1, 3].

## Interpretation of results

The present meta-analysis found a surprisingly large difference in overall complication rate not previously detected in the individual studies. The individual studies mostly analyzed every outcome separately. Each outcome did show a small advantage favoring cement augmentation, but failed to reach statistical significance due to low number of events per outcome. However, when grouped together into a compound endpoint (such as performed in the present meta-analysis), these small differences accumulate to a large difference between treatment groups.

The predominant driving factor behind the difference in overall complication rate was the occurrence of device/fracture related complications. The fact that the risk for re-operation was also higher in the non-augmented group and mostly done for these device/implant related complications signifies their clinical relevance. Four out of 18 (22%) device/fracture related complications required re-intervention in augmented group compared to 21 out of 65 (32%) in the non-augmented group.

The use of cement augmentation does carry an additional risk of cement-specific complications such as leakage into the joint and bone cement implantation syndrome. Leakage of cement into the joint can be prevented by fluoroscopic control using contrast prior to cementation. Minor leakage occurred in some patients in the present study. This, however, did not require any additional intervention. Rare cases of severe bone cement implantation syndrome (BCIS) were only described once in the studies we analyzed [26]. This patient was taken to the intensive care unit and was successfully extubated the day after. Thromboembolic events were rare and occurred slightly more often in the cement group. This however did not reach statistical significance. It is unclear whether this tendency is due to lack of power or whether there truly is no difference. Nevertheless, since the incidence is low, this should not be a reason to refrain from using cement augmentation, to our opinion.

It should be acknowledged that the results found in this meta-analysis are applicable under the condition that the reduction is good. The patients included in the meta-analysis all had adequate reduction and good implant position as can be seen in the baseline tables. Cement augmentation should not be used as a salvage procedure to prevent complications in cases where the surgeon cannot attain adequate reduction. Subgroup analysis on the effect of cement augmentation among patients with inadequate reduction compared to adequate reduction was not possible with the given data. The same applied for subgroup analysis on TAD and implant positioning.

Also, it should be acknowledged that the study population was predominantly female with a mean age around 80 years. Furthermore, they mostly had A2 fractures. Although this represents the typical patient with these fractures according epidemiological studies, we cannot say

to what extent these results can be extrapolated to patients with A1/A3 fractures and/or with a less advanced age (around 65 years) [1].

Cement augmentation carries additional costs related to the use of cement and slightly longer operation duration. However, the cost related to higher complication and re-operation rates and longer hospitalization duration (1.9 days) when no augmentation is used, would most likely outweigh the costs related to cement augmentation. Although, to date, no formal cost-effectiveness analysis exists on this topic, already the costs of 1.9 additional hospitalization days in an academic hospital (1'530 US-Dollar) are much higher compared to the costs of cement (271 US-Dollar) [30].

The present meta-analysis found no difference in weighted effect estimates between randomized clinical trials and observational studies. There is increasing evidence that observational studies yield comparable results as randomized clinical trials in orthopaedic trauma research [20, 31–34]. The potential for confounding, however, should be deemed low when including observational studies. Given the large degree of (baseline) comparability between treatment groups in the present meta-analysis, we considered the potential for confounding acceptably low to allow for inclusion of observational studies in the meta-analysis.

## Limitations

Several limitations should be taken into account. Firstly, there is considerable heterogeneity in half of the outcomes. This heterogeneity is mostly caused by a difference between studies in magnitude of the effect size. All studies do point in the same direction. In other words, it seems fairly certain that cement augmentation is better than no augmentation; to what degree, precisely, suffers from heterogeneity. Secondly, a relatively small number of studies was available for pooled analysis of which particularly the number of observational studies. For some outcomes the comparison of estimates from RCTs and observational studies was based on a sole study in one or both subgroups. Thirdly, although baseline characteristics were comparable across treatment groups (both in the RCTs and observational studies), any residual confounding among observational studies cannot be ruled out. Lastly, in the analysis on overall complication rate, there is a potential for information bias. We are dependent on how detailed studies included in the meta-analysis describe all their complications. Completeness of reporting, therefore, might affect our results.

## Conclusion

This meta-analysis showed that cement augmentation in fixation of trochanteric femur fractures in elderly patients following a low energy trauma leads to fewer complications, re-operations and shorter hospital stay at the expense of a slightly longer operation duration. Cementation specific complications occur rarely and mortality is equal between treatment groups. Radiological and functional results also seem better for cement augmentation.

As the amount of studies included in this meta-analysis is rather small, the results give us an impression on what may be expected of cement augmentation. It does not yet form hard evidence and results mostly apply for patients with advanced age (on average 80 years) with a A2 fracture. To what extent the results can be extrapolated to A1/A3 fractures, remains to be seen. This meta-analysis also underlines the value of including observational studies in meta-analyses.

## Supporting information

**S1 Checklist. PRISMA 2009 checklist.**
(DOC)

**S1 File.**
(DOCX)

## Author Contributions

**Conceptualization:** Ingmar F. Rompen, Ralf Baumgaertner, Bryan J. M. van de Wall.

**Data curation:** Ingmar F. Rompen, Bryan J. M. van de Wall.

**Formal analysis:** Ingmar F. Rompen, Bryan J. M. van de Wall.

**Investigation:** Ingmar F. Rompen, Frank J. P. Beeres, Bryan J. M. van de Wall.

**Methodology:** Ingmar F. Rompen, Bryan J. M. van de Wall.

**Project administration:** Ingmar F. Rompen, Filippo Migliorini.

**Software:** Ingmar F. Rompen, Sven Nebelung.

**Supervision:** Matthias Knobe, Frank J. P. Beeres, Bryan J. M. van de Wall.

**Validation:** Bryan J. M. van de Wall.

**Visualization:** Ingmar F. Rompen.

**Writing – original draft:** Ingmar F. Rompen.

**Writing – review & editing:** Ingmar F. Rompen, Matthias Knobe, Bjoern-Christian Link, Frank J. P. Beeres, Ralf Baumgaertner, Nadine Diwersi, Filippo Migliorini, Sven Nebelung, Reto Babst, Bryan J. M. van de Wall.

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
