## [Decision Letter · Decision Letter 0]

5 Mar 2021

PONE-D-21-00650

Cement augmentation for trochanteric femur fractures: a meta-analysis and systematic review of randomized clinical trials and observational studies Rompen et al.

PLOS ONE

Dear Dr. Rompen,

Thank you for submitting your manuscript to PLOS ONE. After careful consideration, we feel that it has merit but does not fully meet PLOS ONE’s publication criteria as it currently stands. Therefore, we invite you to submit a revised version of the manuscript that addresses the points raised during the review process.

It is indeed a difficult task to perform a metaanalysis. Usually it is a mixture of different studies, often like comparing apples to oranges. Furthermore, reviews are not very welcome in PLOS ONE. Please, omit this item in your title, e.g.: Cement augmentation for trochanteric femur fractures: a meta-analysis  of randomized clinical trials and observational studies.

Two of the three invited reviewers are very familiar with the topic in question. Although they recommended to reject the manuscript, I feel the topic is too important to fail. We would be happy to get a revised version and are convinced, that the major criticisms of the reviewers help to make the manuscript even better.

We look forward to receiving your revised manuscript.

Kind regards,

Hans-Peter Simmen, M.D., Professor of Surgery

Academic Editor

PLOS ONE

Journal Requirements:

2. We noticed you have some minor occurrence of overlapping text with the following previous publications, which needs to be addressed:

- https://www.injuryjournal.com/article/S0020-1383(20)30939-6/fulltext

In your revision ensure you cite all your sources (including your own works), and quote or rephrase any duplicated text outside the methods section.

Further consideration is dependent on these concerns being addressed.

3. During your revisions, please confirm whether the wording in the title is correct and update it in the manuscript file and online submission information if needed. Specifically, it is not necessary to include "Rompen et al." in the online submission form.

4. Please upload a copy of Figures 7 and 8, to which you refer in your text on page 13. If the figure is no longer to be included as part of the submission please remove all reference to it within the text (or amend the text as appropriate).

5. Please include captions for your Supporting Information files at the end of your manuscript, and update any in-text citations to match accordingly. Please see our Supporting Information guidelines for more information: http://journals.plos.org/plosone/s/supporting-information

Reviewers' comments:

Reviewer's Responses to Questions

**Comments to the Author**

1. Is the manuscript technically sound, and do the data support the conclusions?

Reviewer #1: Yes

Reviewer #2: No

Reviewer #3: No

2. Has the statistical analysis been performed appropriately and rigorously? 

Reviewer #1: I Don't Know

Reviewer #2: No

Reviewer #3: Yes

3. Have the authors made all data underlying the findings in their manuscript fully available?

Reviewer #1: Yes

Reviewer #2: Yes

Reviewer #3: Yes

4. Is the manuscript presented in an intelligible fashion and written in standard English?

Reviewer #1: Yes

Reviewer #2: Yes

Reviewer #3: Yes

5. Review Comments to the Author

Reviewer #1: This is a very interesting study which adds valuable and new information to an ongoing discussion. The methods seem to be appropriate as do the results. The derived conclusions are sound. The English is flawless.

I only have minor questions/remarks:

Methods: Norian SRS is mentioned as a device used for fixation. This is misleading.

Discussion Comparison to the literature: You mention an anlalysis comparing cephalomedullary devices to SHS. Where can this analysis be found in your results section? Could you find any differences?

Reviewer #2: Review of the manuscript “Cement augmentation for trochanteric femur fractures: a metaanalysis and systematic review of randomized clinical trials and observational studies.”

The article needs major revision before being published.

The theme of research is quite new, therefore data and studies already published are rare.

In consequence the author cited only 7 articles, 3 of them observational studies (293 OS and 437 RCT in total) which is a too low number to produce an effective metaanalysis.

Cementation in proximal fractures of the femur in elderly is also ethically still an open argument. Observational studies should not be used to make definitive statements of fact about safety, efficacy, or effectiveness of practice. The implementation of these articles in a controversial and new theme of traumatology must be seen very critical. The preventive character of cementation is certainly an interesting new theme in geriatric traumatology but needs more and better research of data.

Second, the risk of bias in including studies is very high and not critically described by the author: high difference in included male/ female patients, different materials for reinforcement ( PMMA/ Calciumphosphate) with different biomechanical criteria, different material and placement for osteosynthesis, very low number of included A1 and A3 fractures. All OS included were talking about intramedullary devices only.

The author gives a general recommendation for cement augmentation in elderly patients over 65. The reviewed articles included 258 augmented A2 fractures and only 56 A3 and 26 A1 fractures. In one of the RCT AO classification was even missing. All patients in all studies were over 80.

The cited articles about cementation effectiveness are only about cadaver studies and in a foam model.

The quality of the evidence of this study to support a specific recommendation is too low.

Finally, I found no PRISMA-P2015 protocol and no PROSPERO registration number for the prospective register of systemic reviews.

Reviewer #3: The primary goal of cement augmentation in osteosynthesis is to prevent implant failure caused by loosening of the implant in very osteoporotic bone (e.g. cut out or displacement of the cephalo-medullary screw). The authors present a comprehensive metaanalysis of 7 publications that compares cemented with uncementeted fixation in the treatment of pertrochanteric fractures. They claim high evidence for their metaanalysis based of the number of pooled patients (n=730) and quality of included studies . Although most studies fail to show clear superiority of cement augmentation, the authors believe that they can show the advantage by comparing the patient groups of 7 studies (n=369 cemented and n=361 uncemented). Considering the high number of surgical procedures this would have a high socio-economical impact.

Critical comment:

The authors perform a comprehensive systematic review of the literature that fulfils high quality demands. But there is a major concern with respect to the data analysed in this systematic review. Among the 7 studies there is a high inhomogeneity with respect to study designs (4 RCTs and 3 observational studies), type of fractures (some include stable fractures some do not) and fixation devices (2 studies analyse the SHS, 5 the intramedullary nail).

The primary outcome parameter of this systematic review was the overall complication rate. But only 2 RCT’s and 3 observational studies could be analysed with respect to this parameter. Out of these 5 studies 3 (Kim, Lee and Yee) give the same numbers/rates for overall and for device related complications (Fig. 2 and Forrest plot Fig. 1). That means that these 3 did not report on overall complications.

With respect to the quality of the studies only one RCT (Kammelander) remains that reports on overall complications. In this study no difference between the cemented and the non-cemented group was seen.

The authors try to explain the benefit of cementation with a lower rate of implant related complications (6% vs. 19%). Only 5 out of 7 studies could be analysed with respect to this parameter (3 observational studies and 2 RCTs). Among these 5 studies 1 RCT reports on SHS in unstable fractures. Since it is evident from the literature that SHS gives worse results for the treatment of unstable fractures in comparison to intramedullary devices, SHS fixation cannot be compared to intramedullary fixation. Without the SHS study the implant related complication rate in the uncemented group would drop from 19 to 11% (still 6% in the cemented group). With respect to the quality of the studies only one RCT (Kammelander) remains that reports on device related complications. In this study no difference between the cemented and the non-cemented group was seen.

According to the conclusion of this systematic review all pertrochanteric fractures would have to be treated with cement augmentation. This is cannot be demanded because of the weak data of the review and it cannot especially be demanded for all types of fractures. Because nearly all studies analysed in this review have reported on unstable fractures (A2 and A3).

With respect to the promising technology of cement fixation of screws in osteoporotic bone it still has to be defined more accurately in which circumstances cement augmentation should be recommended (e.g. grade of fracture instability, grade of osteoporosis), but a general recommendation for all fractures cannot be given according to this review.

On page 4 the authors state that a formal meta-analysis on this topic was not published before. But in the discussion on page 15 they admit that there was one systematic review on that topic published before.

6. PLOS authors have the option to publish the peer review history of their article (what does this mean?). If published, this will include your full peer review and any attached files.

Reviewer #1: No

Reviewer #2: No

Reviewer #3: No

---

## [Author Response · Author response to Decision Letter 0]

28 Apr 2021

Response to Reviewers comments

PONE-D-21-00650

Cement augmentation for trochanteric femur fractures: a meta-analysis of randomized clinical trials and observational studies 

Dear Professor Simmen 

We feel honored that you give us the opportunity to revise our manuscript. We would like to respond to the reviewers comments as follows:

1. We noticed you have some minor occurrence of overlapping text with the following previous publications, which needs to be addressed: https://www.injuryjournal.com/article/S0020-1383(20)30939-6/fulltext

Answer: We made several meta-analyses applying the same methodology including the cited article. For this reason some overlap was expected. 

We changed the methods section of the current manuscript so the overlap is less obvious and added a reference referring to the use of the same methodology of previous meta-analyses produced by our study group.

2. During your revisions, please confirm whether the wording in the title is correct and update it in the manuscript file and online submission information if needed. Specifically, it is not necessary to include "Rompen et al." in the online submission form.

Answer: We changed the format and checked the wording

3. Please upload a copy of Figures 7 and 8, to which you refer in your text on page 13. If the figure is no longer to be included as part of the submission please remove all reference to it within the text (or amend the text as appropriate).

Answer: Figures 8 and 9 are a part of the supplementary material. We changed this in the manuscript so it is more clear. 

Answer: We changed it as requested.

Reviewer #1: This is a very interesting study which adds valuable and new information to an ongoing discussion. The methods seem to be appropriate as do the results. The derived conclusions are sound. The English is flawless.

I only have minor questions/remarks:

1. Methods: Norian SRS is mentioned as a device used for fixation. This is misleading.

Answer: We deleted this in the revised manuscript.

2. Discussion Comparison to the literature: You mention an analysis comparing cephalomedullary devices to SHS. Where can this analysis be found in your results section? Could you find any differences?

Answer: This is presented in the sensitivity analysis. This could only be done for the primary outcome (overall complications). The results are also described in table 2. 

Reviewer #2: Review of the manuscript “Cement augmentation for trochanteric femur fractures: a metaanalysis and systematic review of randomized clinical trials and observational studies.”

The article needs major revision before being published.

1. The theme of research is quite new, therefore data and studies already published are rare.

In consequence the author cited only 7 articles, 3 of them observational studies (293 OS and 437 RCT in total) which is a too low number to produce an effective metaanalysis.

Answer: The seven included studies currently represent the only available data for use in meta-analyses. Indeed, this is a limitation which we addressed in the limitation section. The results give us an impression on what to expect but strong conclusion cannot be drawn. We also underlined this in the conclusion.

2. Cementation in proximal fractures of the femur in elderly is also ethically still an open argument. Observational studies should not be used to make definitive statements of fact about safety, efficacy, or effectiveness of practice. The implementation of these articles in a controversial and new theme of traumatology must be seen very critical. The preventive character of cementation is certainly an interesting new theme in geriatric traumatology but needs more and better research of data.

Answer: The use of observational studies is indeed a point of debate. However, almost all previous meta-analyses that included observational studies and performed subgroup analysis stratified for study design (observational studies versus randomised clinical trials), found that the pooled estimates of observational studies were identical to those obtained from randomised clinical trials in trauma research. For this reason, we included observational data as well as randomised trials. We addressed this topic in the discussion and added references of previous meta-analyses demonstrating this. We also aknowledge that readers should be careful as the number of studies available for testing this assumption within this paper is limited. 

3. Second, the risk of bias in including studies is very high and not critically described by the author: high difference in included male/ female patients, different materials for reinforcement (PMMA/ Calciumphosphate) with different biomechanical criteria, different material and placement for osteosynthesis, very low number of included A1 and A3 fractures. All OS included were talking about intramedullary devices only.

Answer: We do not fully understand the comment. The baseline characteristics are equally distributed among treatment groups; therefore, the risk of bias for the measured characteristics is low. We think the reviewer is referring to the heterogeneity of the study population. Indeed it is a mix of patients with a specific gender distribution (in our opinion representative of daily practice as the majority of patients with proximal femur fractures are indeed female) that were treated with different types of cement and osteosynthesis devices and different quality of reduction/material placement. The main question is to what extent these aspects contributed individually to the found differences between the treatments. We attempted to investigate this via a sensitivity analysis (see table 2), however lack the power to study this in detail.

We feel that this is important to describe in the discussion section. We hope the added section and our answer satisfies the comments of the reviewer. 

4. The author gives a general recommendation for cement augmentation in elderly patients over 65. The reviewed articles included 258 augmented A2 fractures and only 56 A3 and 26 A1 fractures. In one of the RCT AO classification was even missing. All patients in all studies were over 80.

Answer: The reason for recommending augmentation for patients older than 65 years is because all included studies shared the same inclusion criteria, namely, that only patients older than 65 years were included. The fact that the mean age of the study population was around 80 years, has consequences for generalizability of results. The same applies for the fact that predominantly A2 fractures were included. All in all, results found in this study mostly apply for patients similar to the patients included in this meta-analysis being 80 year old females with an A2 fracture. We added a sentence to the recommendation that readers should be aware that results mostly apply to a specific population and to lesser extent for the entire spectrum of proximal femur fractures.

5. The cited articles about cementation effectiveness are only about cadaver studies and in a foam model.

The quality of the evidence of this study to support a specific recommendation is too low.

Answer: Cited articles in the introduction are indeed about cadaveric studies and foam models. We would like to assure that the studies included in the meta-analysis were clinical studies on real patients. 

6. Finally, I found no PRISMA-P2015 protocol and no PROSPERO registration number for the prospective register of systemic reviews.

Answer: This is, regrettably, correct. Publishing a protocol prior to performing a meta-analysis would have contributed to transparency. 

Reviewer #3: The primary goal of cement augmentation in osteosynthesis is to prevent implant failure caused by loosening of the implant in very osteoporotic bone (e.g. cut out or displacement of the cephalo-medullary screw). The authors present a comprehensive metaanalysis of 7 publications that compares cemented with uncementeted fixation in the treatment of pertrochanteric fractures. They claim high evidence for their metaanalysis based of the number of pooled patients (n=730) and quality of included studies. Although most studies fail to show clear superiority of cement augmentation, the authors believe that they can show the advantage by comparing the patient groups of 7 studies (n=369 cemented and n=361 uncemented). Considering the high number of surgical procedures this would have a high socio-economical impact.

1. The authors perform a comprehensive systematic review of the literature that fulfils high quality demands. But there is a major concern with respect to the data analysed in this systematic review. Among the 7 studies there is a high inhomogeneity with respect to study designs (4 RCTs and 3 observational studies), type of fractures (some include stable fractures some do not) and fixation devices (2 studies analyse the SHS, 5 the intramedullary nail).

Answer: We would like to refer to our answer on comment 3 of reviewer 2 as he pointed out the same issue. 

2. The primary outcome parameter of this systematic review was the overall complication rate. But only 2 RCT’s and 3 observational studies could be analysed with respect to this parameter. Out of these 5 studies 3 (Kim, Lee and Yee) give the same numbers/rates for overall and for device related complications (Fig. 2 and Forrest plot Fig. 1). That means that these 3 did not report on overall complications.

With respect to the quality of the studies only one RCT (Kammelander) remains that reports on overall complications. In this study no difference between the cemented and the non-cemented group was seen.

Answer: This was indeed something we also struggled with during the analysis phase. We are highly dependent how detailed studies report their outcomes. For this reason we splitted the analysis of complications in overall, device related and systematic complications so readers may extract the information that they need. We feel that this is a good point that readers should be aware of and added this to the limitation section.

3. The authors try to explain the benefit of cementation with a lower rate of implant related complications (6% vs. 19%). Only 5 out of 7 studies could be analysed with respect to this parameter (3 observational studies and 2 RCTs). Among these 5 studies 1 RCT reports on SHS in unstable fractures. Since it is evident from the literature that SHS gives worse results for the treatment of unstable fractures in comparison to intramedullary devices, SHS fixation cannot be compared to intramedullary fixation. Without the SHS study the implant related complication rate in the uncemented group would drop from 19 to 11% (still 6% in the cemented group). With respect to the quality of the studies only one RCT (Kammelander) remains that reports on device related complications. In this study no difference between the cemented and the non-cemented group was seen.

Answer: As previously described in the answer to point 3 of reviewer 2, the most important factor for limiting risk of confounding when comparing two treatment regimens is that study population characteristics are equally contributed between treatment groups. Including both studies that used SHS or intramedullary nailing into the meta-analysis has consequences for generalizability of results. We attempted to study how much each factor contributed to the occurrence of significant differences between cement and no cement augmentation in the sensitivity analysis including SHS versus intra-medullary nailing. However, logically, we are limited by the data available for this meta-analysis. 

On the other hand, combining studies into a meta-analysis increases sample size and with it the ability to detect differences not previously found in individual studies. This might explain why Kammerlander et al did not find a difference and the present meta-analysis did. 

4. According to the conclusion of this systematic review all pertrochanteric fractures would have to be treated with cement augmentation. This is cannot be demanded because of the weak data of the review and it cannot especially be demanded for all types of fractures. Because nearly all studies analysed in this review have reported on unstable fractures (A2 and A3). With respect to the promising technology of cement fixation of screws in osteoporotic bone it still has to be defined more accurately in which circumstances cement augmentation should be recommended (e.g. grade of fracture instability, grade of osteoporosis), but a general recommendation for all fractures cannot be given according to this review.

Answer: We fully agree with the reviewer. This was also pointed out by the other reviewers. Therefore we changed the conclusion explaining that results a mostly applicable for patients with A2/3 fractures with an average age of 80 years. Whether the results apply to the full spectrum of proximal femur fractures has yet to be defined.

On page 4 the authors state that a formal meta-analysis on this topic was not published before. But in the discussion on page 15 they admit that there was one systematic review on that topic published before.

Answer: This was a qualitative systematic review without a quantitative pooled analysis, such as performed in the present study. For this reason, we described it in this fashion. We felt that the comparison of the systematic review to our meta-analysis was best addressed in the discussion section. 

We hope having addressed your remarks and concern appropriately. We are very happy to receive further comments that can improve the quality of our research. 

Sincerely yours,

Ingmar Rompen, Bryan van de Wall and research team

---

## [Editor Report · Decision Letter 1]

5 May 2021

Cement augmentation for trochanteric femur fractures: a meta-analysis of randomized clinical trials and observational studies

PONE-D-21-00650R1

Dear Dr. Rompen,

We’re pleased to inform you that your manuscript has been judged scientifically suitable for publication and will be formally accepted for publication once it meets all outstanding technical requirements.

Kind regards,

Hans-Peter Simmen, M.D., Professor of Surgery

Academic Editor

PLOS ONE
---

## [Editor Report · Acceptance letter]

6 May 2021

PONE-D-21-00650R1 

Cement augmentation for trochanteric femur fractures: a meta-analysis of randomized clinical trials and observational studies. 

Dear Dr. Rompen:

I'm pleased to inform you that your manuscript has been deemed suitable for publication in PLOS ONE. Congratulations! Your manuscript is now with our production department. 

Kind regards, 

on behalf of

Dr. Hans-Peter Simmen 

Academic Editor

PLOS ONE